

# Marine cyanolichens from different littoral zones are associated with distinct bacterial communities

Nyree J. West[1], Delphine Parrot[2,5], Claire Fayet[1], Martin Grube[3], Sophie Tomasi[2] and Marcelino T. Suzuki[4]

[1] Observatoire Océanologique de Banyuls sur mer, Sorbonne Université, CNRS, Banyuls sur mer, France
[2] Univ Rennes, CNRS, ISCR—UMR 6226, Rennes, France
[3] Institute of Plant Sciences, University of Graz, Graz, Austria
[4] Laboratoire de Biodiversité et Biotechnologies Microbiennes (LBBM), Sorbonne Université, CNRS, Banyuls sur mer, France
[5] Current address: GEOMAR Helmholtz Centre for Ocean Research Kiel, Research Unit Marine Natural Products Chemistry, GEOMAR Centre for Marine Biotechnology, Kiel, Germany

Corresponding author
Nyree J. West,
nyree.west@obs-banyuls.fr

## ABSTRACT

The microbial diversity and function of terrestrial lichens have been well studied, but knowledge about the non-photosynthetic bacteria associated with marine lichens is still scarce. 16S rRNA gene Illumina sequencing was used to assess the culture-independent bacterial diversity in the strictly marine cyanolichen species *Lichina pygmaea* and *Lichina confinis*, and the maritime chlorolichen species *Xanthoria aureola* which occupy different areas on the littoral zone. Inland terrestrial cyanolichens from Austria were also analysed as for the marine lichens to examine further the impact of habitat/lichen species on the associated bacterial communities. The *L. confinis* and *L. pygmaea* communities were significantly different from those of the maritime *Xanthoria aureola* lichen found higher up on the littoral zone and these latter communities were more similar to those of the inland terrestrial lichens. The strictly marine lichens were dominated by the Bacteroidetes phylum accounting for 50% of the sequences, whereas Alphaproteobacteria, notably *Sphingomonas*, dominated the maritime and the inland terrestrial lichens. Bacterial communities associated with the two *Lichina* species were significantly different sharing only 33 core OTUs, half of which were affiliated to the Bacteroidetes genera *Rubricoccus*, *Tunicatimonas* and *Lewinella*, suggesting an important role of these species in the marine *Lichina* lichen symbiosis. Marine cyanolichens showed a higher abundance of OTUs likely affiliated to moderately thermophilic and/or radiation resistant bacteria belonging to the Phyla Chloroflexi, Thermi, and the families Rhodothermaceae and Rubrobacteraceae when compared to those of inland terrestrial lichens. This most likely reflects the exposed and highly variable conditions to which they are subjected daily.

# INTRODUCTION

The lichen symbiosis, commonly recognised as a partnership of a fungus (mycobiont), and a photosynthetic partner (photobiont) arose with the conquest of land in the lower

Devonian, according to the first clear fossil evidence (*Honegger, Edwards & Axe, 2013*). The enclosure of the photobiont partner by protective layers of the fungal partner gave rise to a new morphological structure. This symbiotic structure is called the lichen thallus, which apparently mediates a high degree of tolerance to desiccation (*Kranner et al., 2008*), and allows many lichens to thrive as poikilohydric organisms in environments characterised by periodic changes in environmental conditions. Therefore, lichens are typically found in habitats where other organisms struggle to persist such as the intertidal belt of coastal rocks, where lichens develop characteristically coloured belts.

Bacteria were already found in the oldest microfossils that can be reliably assigned to lichen thalli (*Honegger, Edwards & Axe, 2013*), an observation that fits well with the observations of bacterial colonisation of extant lichens (*Cardinale et al., 2008*). Recent works suggest that the ubiquitous presence of bacteria in lichen thalli contributes to a more complex functional network beyond the interaction of fungi and algae (*Aschenbrenner et al., 2016*). Bacteria were first cultured from lichens many years ago and were originally supposed to be involved in nitrogen-fixation (*Henkel & Yuzhakova, 1936*). However, due to the low culturability of bacteria from many environments (*Ferguson, Buckley & Palumbo, 1984*) and the tendency of culture methods to select for opportunistic species which rarely dominate in the natural environment (*Eilers, Pernthaler & Amann, 2000*), their diversity was only recently fully recognised.

Culture-independent molecular studies of lichen microbial diversity as well as microscopic observations revealed that bacteria belonging to the Alphaproteobacteria were the dominant microbial group associated with the lichens (*Cardinale, Puglia & Grube, 2006*; *Liba et al., 2006*; *Grube et al., 2009*; *Bjelland et al., 2011*; *Hodkinson et al., 2012*; *Cardinale et al., 2012a*). In these studies, high abundances of Alphaproteobacteria were generally observed on the surface structures of the lichen thalli although some were observed in the fungal hyphae. More recently, the application of high-throughput sequencing techniques to lichen bacterial community analysis confirmed that the composition of the lichen bacterial communities could be more influenced by the mycobiont species than by their sample site (*Bates et al., 2011*), by the photobiont (*Hodkinson et al., 2012*), and also by the position in the lichen thallus (*Mushegian et al., 2011*).

Lichens exhibit clear specificity for substrate and microhabitat conditions, and a clear example of habitat specialisation can be observed for marine lichens, which show vertical zonation in four characteristic belts along rocky coastlines (also known as sublittoral, littoral, supra-littoral and terrestrial zones (*Fletcher, 1973a*; *Fletcher, 1973b*)). The common littoral lichen *Lichina pygmaea* is immersed for several hours each day whereas *Lichina confinis* occurs higher up in the littoral zone, where it is perpetually subjected to splashing and sea-spray and submerged only during short periods of high tides. *Xanthoria* spp. can also occur in this zone and in the xeric supralittoral zone, which is exposed to sea-spray during storms but not submerged in seawater. Therefore, marine lichen species certainly experience different levels of stress, ranging from direct sunlight exposure, temperature, salinity and wind variation according to their position in the littoral zone (*Delmail et al., 2013*).

The genus *Lichina* belongs to the class Lichinomycetes and is composed of both strictly marine and non-marine species (*Schultz, 2017*). The marine species *L. confinis* and *L. pygmaea* harbour cyanobacterial photobionts closely related to strains of the genus *Rivularia* (*Ortiz-Álvarez et al., 2015*). Even though the two lichen species above show a similar distribution range, their cyanobionts belong to separate groups that do not overlap at the OTU or even at the haplotype level (*Ortiz-Álvarez et al., 2015*). Apart from the cyanobacterial photobiont, the composition of the bacteria associated with marine lichens is poorly studied when compared to those of inland lichens. So far, only *Hydropunctaria maura*, which forms the black belt on littoral zones, was included in a culture-independent study (*Bjelland et al., 2011*). Bacterial communities of this lichen were different from inland terrestrial lichens, with higher relative abundances of Actinobacteria, Bacteroidetes, Deinococcus, and Chloroflexi. A recent study of Icelandic marine lichens focused on culturable bacteria, but also revealed by fingerprinting (DGGE) analysis of 16S rRNA genes that the bacterial communities were different among different marine lichen species (*Sigurbjornsdottir et al., 2014*).

Such differences could be of interest for bioprospecting approaches since lichens are known to be a rich source of natural products (*Boustie & Grube, 2005*; *Parrot et al., 2016*). However, lichen-associated bacteria have only recently been discovered as an additional contributor to the lichen chemical diversity, and even though only few lichen-associated bacteria have been studied carefully in this respect, many of them produce secondary metabolites. Some of their compounds have pronounced bioactive properties (see *Suzuki et al., 2016* for a review). Nevertheless, the role of these compounds in the symbiosis is not well understood at this stage. So far, bioprospecting of the marine lichens *L. pygmaea* and *L. confinis* (and *Roccella fuciformis*) was carried out by a cultivation-based approach targeting Actinobacteria that led to the isolation of 247 bacterial species including 51 different Actinobacteria (*Parrot et al., 2015*).

To address the paucity of knowledge on the total diversity and variation of bacteria associated with marine lichens and to provide a base for future bioprospecting efforts, the uncultured microbial communities of the cyanolichen species *L. confinis* and *L. pygmaea* and the chlorolichen species *X. aureola* and *Xanthoria parietina* were characterised by Illumina sequencing of 16S rRNA gene fragments. In addition, other inland terrestrial cyanolichen species were characterised in parallel to allow us to address the following questions: (i) what are the major bacterial groups associated with the marine and maritime lichens occupying different heights on the littoral zone; (ii) how does the culturable fraction of bacteria from *Lichina* spp. and *Lathagrium auriforme* compare to the total bacterial community associated with these species; and (iii) do inland terrestrial cyanolichens share a significant proportion of their symbionts with marine cyanolichens?

## MATERIALS & METHODS

### Sampling design and sample collection

The marine cyanolichens *Lichina pygmaea* and *Lichina confinis* were collected on 16 October 2013 from the Houssaye Point, Erquy on the coast of Brittany, France (Table 1, Fig. 1).

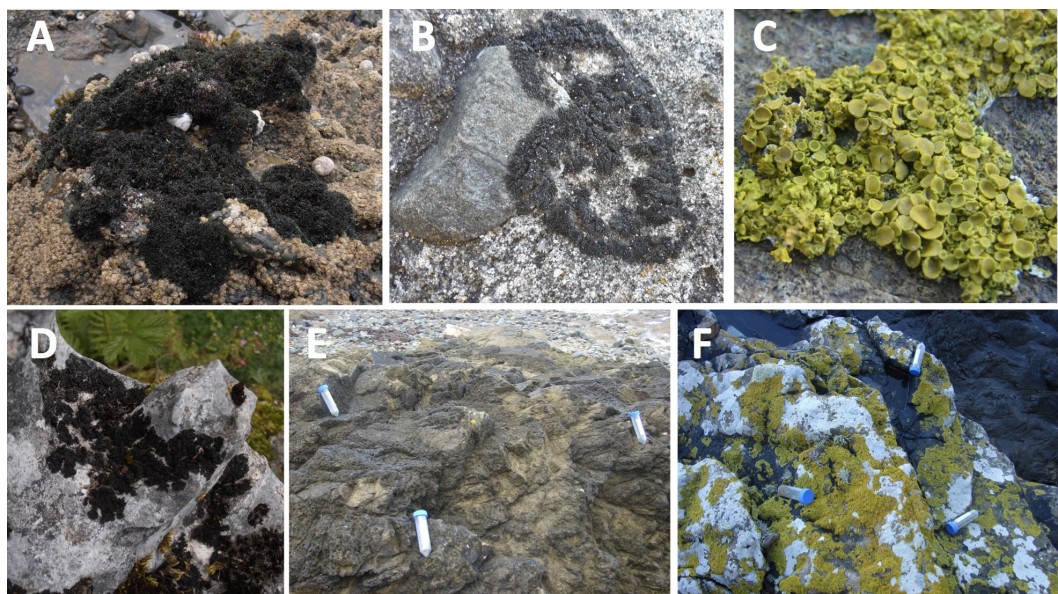

**Figure 1** **Examples of lichens collected in this study.** (A) *Lichina pygmaea* (B) *Lichina confinis* (C) *Xanthoria aureola* (Brittany) (D) *Lathagrium auriforme* (E) *L. confinis* sample cluster (F) *X. aureola* sample cluster. Photos by Delphine Parrot.

**Table 1** **Lichen species analysed in this study.** The different lichen species sampled in this study are presented with details on their corresponding mycobiont and photobiont partners, their habitat and the location of the sampling sites.

| Lichen species | Mycobiont | Photobiont | Habitat | Date | Location |
|---|---|---|---|---|---|
| *Lichina confinis* | *Lichinomycetes, Lichinales, Lichinaceae* | Cyanobacteria: *Rivularia* sp | Upper eulittoral | 16.10.13 | Erquy, France (Atlantic Ocean) |
| *Lichina pygmaea* | *Lichinomycetes, Lichinales, Lichinaceae* | Cyanobacteria: *Rivularia* sp | Lower eulittoral | 16.10.13 | Erquy, France (Atlantic Ocean) |
| *Xanthoria aureola* | *Lecanoromycetes, Teloschistales, Teloschistaceae* | Green alga: *Trebouxia* sp | Mesic-supralittoral | 16.10.13 | Erquy, France (Atlantic Ocean) |
| *Xanthoria parietina* | *Lecanoromycetes, Teloschistales, Teloschistaceae* | Green alga: *Trebouxia* sp | Xeric-supralittoral | 06.03.14 | Banyuls sur mer, France (Mediterranean Sea) |
| *Lathagrium auriforme* | *Lecanoromycetes, Peltigerales Collemataceae* | Cyanobacteria: *Nostoc* sp. | Terrestrial | 26.11.13 | Kesselfalklamm, Austria |
| *Lathagrium cristatum* | *Lecanoromycetes, Peltigerales Collemataceae* | Cyanobacteria: *Nostoc* sp. | Terrestrial | 26.11.13 | Kesselfalklamm, Austria |
| *Lathagrium fuscovirens* | *Lecanoromycetes, Peltigerales Collemataceae* | Cyanobacteria: *Nostoc* sp. | Terrestrial | 26.11.13 | Kesselfalklamm, Austria |
| *Scytinium lichenoides* | *Lecanoromycetes, Peltigerales Collemataceae* | Cyanobacteria: *Nostoc* sp. | Terrestrial | 26.11.13 | Kesselfalklamm, Austria |

Lichens were sampled in triplicate in a 3 m radius zone that we defined as a sample cluster. Two other sample clusters of triplicates were sampled further along the littoral zone. For *L. confinis*, sample clusters 1 and 2 were 22 m apart and clusters 2 and 3 were 14.4 m apart. For *L. pygmaea*, clusters 1 and 2, and 2 and 3, were both separated by 6 m. Duplicate seawater samples were also taken using sterile 50 ml centrifuge tubes. Similarly, one cluster of three *Xanthoria aureola* samples was also taken from Houssaye Point and another cluster of *Xanthoria parietina* was sampled from the rocky coastline of Banyuls-sur-mer on the Mediterranean Sea (Table 1, Table S1). The inland terrestrial cyanolichens were sampled at Kesselfalklamm, Austria. *Lathagrium auriforme* and *Scytinium lichenoides* were sampled in the shaded humid zone near to a river whereas *Lathagrium cristatum* and *Lathagrium fuscovirens* were collected from rocks in an exposed dry zone (Table 1, Table S1, Fig. 1). Lichens were sampled using a sterile scalpel and gloves, placed in sterile bags and stored at −80 °C.

The morphological identification of the lichens was verified by sequencing the marker genes ITS-5.8S rRNA (*White et al., 1990*) for the marine and maritime lichens and the genes beta-tubulin (*Myllys, Lohtander & Tehler, 2001*) and nuclear LSU rRNA (*Vilgalys & Hester, 1990*; *Otálora et al., 2010*) for the inland terrestrial lichens (Supplementary Information, Table S2).

## DNA extraction

Genomic DNA was extracted using the DNeasy PowerPlant Pro DNA Isolation Kit (Qiagen S.A.S., Courtaboeuf, France). Lysis was achieved by first grinding the lichens (several pieces selected randomly from each thallus amounting to approximately 0.1 g of material) to a powder with liquid nitrogen using a sterile pestle and mortar. The powder was added to the lysis tubes of the kit following the protocol according to the manufacturer's instructions, including a heating step of 65 °C for 10 min before the addition of the RNase A solution. Homogenisation was achieved using a standard vortex equipped with a flat-bed vortex pad. DNA quality was verified by agarose gel electrophoresis and DNA was quantified using the Quant-iT$^{TM}$ PicoGreen® dsDNA Assay Kit (Invitrogen, Thermo-Fisher Scientific Inc., Courtaboeuf, France) according to manufacturer's instructions.

## PCR amplification of 16S rRNA genes and illumina sequencing

Bacterial 16S rRNA gene fragments were amplified using the primer pair 341F (CCTACGGGNGGCWGCAG) and 805R (GACTACHVGGGTATCTAATCC) which show the best combination of domain and phylum coverage (*Klindworth et al., 2012*). The 341F primers were tagged at the 5′end with different 7 bp tags for each sample (Table S3) that were chosen from a set of tags designed to be robust to substitution, deletion and insertion errors incurred in massively parallel sequencing (*Faircloth & Glenn, 2012*). We also included in parallel a control consisting of DNA from a synthetic mock community (Mock) of 20 bacterial species containing equimolar (even) rRNA operon counts (HM-782D; Genomic DNA from Microbial Mock Community B, Even, Low Concentration, BEI Resources, Manassas, VA. This standard is now obtainable from LGC Standards S.a.r.l., Molsheim, France; reference ATCC® MSA-1002$^{TM}$). Between 1-10 ng

of each DNA sample (or 1 μl of Mock DNA) were amplified in duplicate 10 μl reactions containing 1X KAPA 2G Fast Ready Mix (Merck, Darmstadt, Germany) and 0.5 μM of each primer. The PCR cycling conditions were 95 °C for 3 min followed by 25 cycles of 95 °C for 15s, 55 °C for 15s and 72 °C for 2s, and a final extension of 72 °C for 30s. Duplicate reactions were pooled, and PCR amplification verified by gel electrophoresis. To normalise the samples before pooling and sequencing, the Sequalprep Normalization Plate (96) kit (Invitrogen) was used according to manufacturer's instructions. After binding, washing and elution, the 37 environmental samples and the mock DNA sample were pooled into one tube. The clean-up of the PCR amplicon pool was achieved with the Wizard SV Gel and PCR Clean-Up System (Promega, Charbonnières-les-Bains, France) according to manufacturer's instructions with elution in 30 μl of molecular biology grade water (Merck). DNA was quantified with the Quant-iT PicoGreen dsDNA Assay kit (Invitrogen) according to manufacturer's instructions. Approximately 350 ng DNA was pooled with 700 ng of barcoded PCR products from a different project (sequencing the 16S rRNA genes of marine bacterioplankton) and sent out to the sequencing company Fasteris (Geneva, Switzerland) for library preparation and sequencing. Library preparation involved ligation on PCR using the TruSeq DNA Sample Preparation Kit (Illumina, San Diego, CA, USA) according to manufacturer's instructions except that five PCR cycles were used instead of 10 cycles. The library was sequenced on one Illumina MiSeq run using the $2 \times 300$ bp protocol with MiSeq version 3.0 chemistry and the basecalling pipeline MiSeq Control Software 2.4.1.3, RTA 1.18.54.0 and CASAVA-1.8.2. The error rate was measured by spiking the library with about 0.5% of a PhiX library and mapping the reads onto the PhiX reference genome.

## Sequence pre-processing

Paired-end Illumina sequence data was processed with a custom pipeline (Supplementary Information) using mainly the USEARCH-64 version 8 package (Drive5; Tiburon, CA, USA) with some commands from QIIME version 1.9.1 (*Caporaso et al., 2010*) and *mothur* version 1.33.3 (*Schloss et al., 2009*). Analysis of the Mock data set first allowed us to calibrate the different steps of the sequence processing pipeline ensuring that the assigned OTUs reflected as closely as possible the defined composition of the sample and allowed us to reveal potential sequencing artifacts (errors, contamination). To minimise errors and reduce over-inflation of diversity, the following criteria were chosen in the pre-processing steps; (a) zero mismatches allowed in the overlapping region when merging the paired end reads, (b) quality filtering was carried out after merging using a stringent expected error of 1.0 (*Edgar & Flyvbjerg, 2015*), (c) zero mismatches allowed in the barcode when demultiplexing, (d) exact matches to both primers required, (e) removal of singleton sequences that are expected to have errors and that increase the number of spurious OTUs (*Edgar, 2013*). Sequence data was submitted to the European Nucleotide Archive under the study number: PRJEB23513.

## Sequence data analysis

OTUs were defined with the UPARSE algorithm, part of the USEARCH-64 package, which clusters sequences with >97% identity to OTU centroid sequences (representative

sequences that were selected based on their rank of read abundance after dereplication) and that simultaneously removes chimeras (*Edgar, 2013*). Subsequent analysis steps were performed in QIIME or MACQIIME version 1.9.1. OTU taxonomy was assigned with the RDP classifier using the GreenGenes OTU database (gg_13_8_otus) defined at 97% identity. For the lichen dataset, the OTU table was filtered to remove eukaryotic, archaeal and organelle sequences (chloroplast, mitochondria). Due to the presence of the lichen photobiont, a large number of cyanobacterial sequences were recovered. The OTU table was filtered to separate non-cyanobacteria sequences from the cyanobacteria sequences. The representative sequences were aligned using the PYNAST aligner in QIIME and the 13_8 GreenGenes alignment (Supplementary Information), and the OTU table was further pruned by removing non-aligning OTUs. Data exploration and analysis was carried out with QIIME (see Supplementary Information) and with the packages Phyloseq (*McMurdie & Holmes, 2013*) and vegan (*Oksanen et al., 2013*) in R (*R Core Team, 2015*). One-way ANOSIM tests were performed using the vegan package in R to determine significant differences between groups. To examine the patterns of phylogenetic relatedness of the lichen-associated bacterial communities, the mean pairwise distance (MPD) and the mean nearest taxon distance (MNTD) were calculated using the Picante package in R (*Kembel et al., 2010*). Whereas MPD calculates the mean pairwise distance between all OTUs in each community, MNTD calculates the mean distance between each OTU in the community and its closest relative. To test the significance of the phylogenetic community structure observed, the standardised effect size (SES) was then calculated for the MPD and MNTD to compare the observed pattern to that of a null model of phylogeny. We used the null model of randomly shuffling the tip labels. $SES_{(MPD)}$ and $SES_{(MNTD)}$ are equivalent to $-1$ times the Net Relatedness Index (NRI) and the Nearest Taxon Index (NTI) respectively, reported previously (*Webb, Ackerly & Kembel, 2008*).

Since analysis of the Mock data revealed the presence of contaminant and/or sequences from a library from another project, potentially due to cross-talk during the Illumina sequencing, OTUs exhibiting a low uniform abundance amongst all the samples were filtered out by calculating between-sample variance on a relative abundance OTU table and removing those OTUs with a variance of $<1 \times 10^{-6}$.

## Comparison of sequences with cultured species, and the NBCI database

The 16S rRNA gene sequences of 247 bacterial strains previously isolated from the lichens *L. pygmaea*, *L. confinis,* and *L. auriforme* (*Parrot et al., 2015*) were compared to the sequences from this study by *blastn* (*Altschul et al., 1997*) queries against a database of the representative OTU sequences in the current study using the standalone version of NCBI blast. The bacterial strains were isolated from the lichens in France and Austria 18 months and 12 months respectively before those sampled in this study, but they were sampled at exactly the same sites. Note that in the previous study (*Parrot et al., 2015*), the lichen *L. auriforme* was cited with its previous name; *C. auriforme* (*Collema auriforme*).

In addition, the sequences of the top 30 OTUS from the strict marine or maritime-inland terrestrial lichens were subjected to *blastn* queries against the NCBI Genbank nt database.

## RESULTS

### Control of sequence quality and data treatment

The mock community sequences were parsed out and analysed separately from the other sequences obtained from the lichen and seawater samples. Analysis of the 20-species mock community allowed us to check for potential contamination and also to refine our bioinformatic analysis pipeline to obtain the expected number and identity of OTUs. Using the criteria described in the methods and the algorithm UPARSE of the USEARCH package for clustering OTUs at 97% identity, we obtained 19 OTUs (identical to the reference sequence *Edgar, 2013*) which perfectly corresponded to the mock community (the two *Staphylococcus* sp. strains cluster together) plus 17 other OTUs. The 19 mock OTUs accounted for 99.5% of the sequences. Of the remaining 17 OTUs, four were affiliated to the Cyanobacteria class and three to chloroplasts that were the dominant OTUs found respectively in the lichen samples (this study) and in samples from the offshore bacterioplankton study loaded in the same MiSeq run. This is evidence of the sequencing artifact termed "cross-talk" which can be due to index misassignment where reads from one sample are wrongly assigned the indexes from a different sample. The remaining 10 mock OTUs accounted for 0.24% of the reads.

After the different sequence processing steps on the lichen and seawater samples, a total of 509,656 good quality sequences obtained were clustered into 93 cyanobacterial OTUs and 2,519 bacterial OTUs. Examination of the OTU tables of the study samples also revealed cross-talk OTUs as observed in the mock sample that were mainly attributed to chloroplasts and cyanobacteria. This problem was partly resolved by removal of these lineages for subsequent analysis of the heterotrophic bacterial diversity but there were nevertheless some bacterioplankton OTUs present at consistently low abundances in all the lichen samples, which are dominant in the samples from different study (e.g., Pelagibacteraceae). Therefore, for certain analyses, the OTU table was filtered for low variance OTUs which allowed the removal of spurious OTUs. This data treatment would also have removed non-spurious low abundance OTUs but as our analysis was focused on the dominant (most abundant) OTUs and not the rare OTUs, this would not affect our major findings.

As expected for cyanolichens, a high proportion of the sequences were affiliated to the Cyanobacteria lineage (including chloroplasts) and ranged from 30–60% for the marine lichens and from 40–85% for the inland terrestrial lichens (Fig. S1). The cyanobacterial sequences were then removed for further analyses. The number of non-cyanobacteria bacterial sequences per sample ranged from 2,759 for *L. auriforme* to >14,000 sequences for *X. aureola*.

### Bacterial communities associated with lichens in different areas of the littoral zone

For clarity, we use the term "marine lichens" to refer to the two *Lichina* species that are strictly marine, "maritime lichens" to refer to the two *Xanthoria* species and "inland terrestrial lichens" the other lichens isolated from Austria.

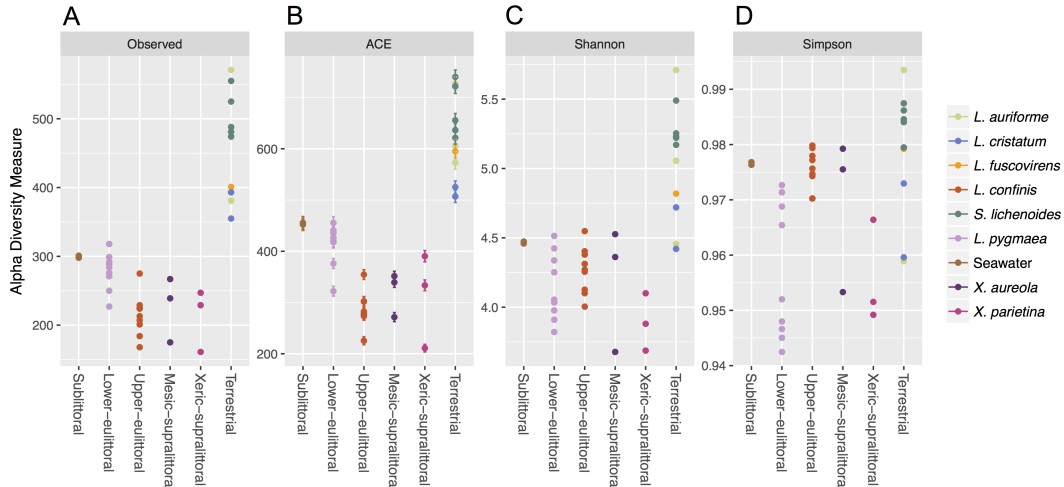

**Figure 2** **Alpha diversity of bacterial communities associated with marine, maritime and inland terrestrial lichens, and in seawater.** The richness ((A) Observed OTUs and (B) Abundance-based Coverage Estimator (ACE) of species richness), and diversity ((C) Shannon diversity and (D) Simpson diversity) metrics for the lichen species or seawater bacterial communities are displayed by position on the littoral zone for the marine and maritime lichens and the lichens from Austria are plotted as the Terrestrial category.

The alpha diversity of the non-cyanobacteria bacterial communities associated with the marine and maritime lichens and in the adjacent seawater was compared using different species diversity and richness indices calculated with a rarefied OTU table (Fig. 2). The marine and maritime lichens showed a significantly lower diversity than the inland terrestrial lichens ($t_{33} = 3.35$, $p = 0.001$ for Inverse Simpson). *L. pygmaea* bacterial communities exhibited higher species richness than those of *L. confinis* (observed OTUs and ACE) but showed lower diversity as indicated by the Shannon and Inverse Simpson diversity metrics.

The taxonomic diversity at the phylum level of the non-cyanobacteria bacteria associated with the different lichens is shown in Fig. 3. Bacteroidetes and Proteobacteria were the dominant phyla in the marine and maritime lichens although the *Lichina* lichens showed a higher relative abundance of Bacteroidetes (>50% of the sequences) and Chloroflexi compared to the *Xanthoria* species. The two chlorolichen *Xanthoria* species collected from either the Mediterranean Sea or Atlantic Ocean coastlines were also associated with the candidate phylum FBP that was either absent or associated at very low abundance in the *Lichina* spp. Interestingly, the phylum Thermi comprising many thermophilic species were predominantly associated with the *Lichina* and *Xanthoria* lichens.

A comparison of the lichen bacterial community beta diversity as assessed by Bray–Curtis dissimilarity of a rarefied OTU table and hierarchical clustering is shown in Fig. 4A. The dendrogram shows a clear separation of the marine cyanolichens from the seawater samples and also from the maritime lichens whose bacterial communities formed a separate cluster with the inland terrestrial cyanolichens. The data were then reanalysed, focusing on the most abundant OTUs associated with the lichens. This involved removing the seawater samples

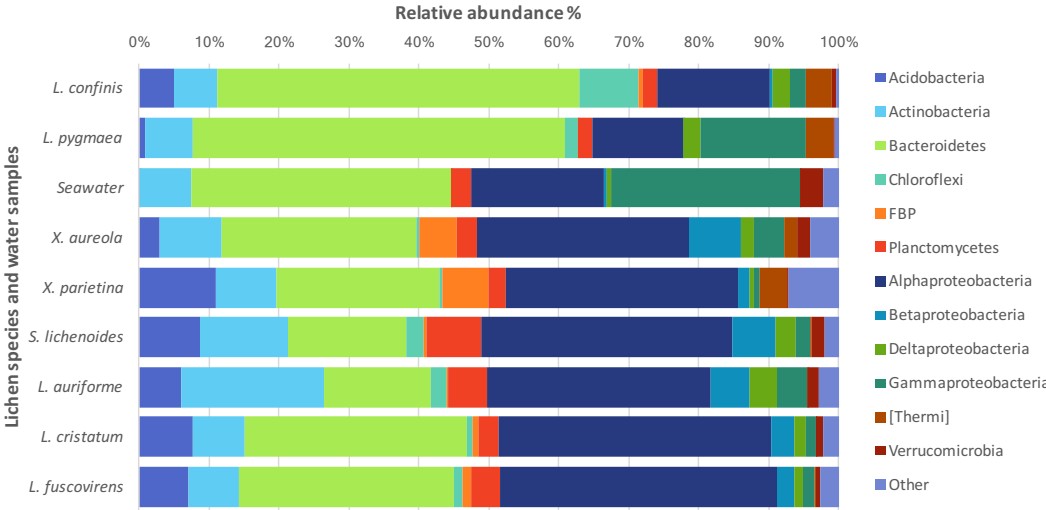

**Figure 3** **Phylogenetic diversity of the bacteria associated with different lichen species or in seawater.**
The relative abundances of the major bacterial phyla and Proteobacteria classes associated with lichens or
in seawater are expressed as percentages of the total sequences. Phyla that accounted for less than 2% of
the total sequences were grouped together as ''Other''.

from the dataset and filtering out low variance OTUs thus reducing the OTU number
from 2,519 to 367. Despite this filtering step, the clustering of the lichen communities of
the reduced dataset (Fig. S2) was similar to that of the full dataset (Fig. 4A). ANOSIM
tests confirmed that the separation of marine versus the other lichen groups, and also the
separation of the *L. confinis* and *L. pygmaea* groups was highly significant ($R = 1$, $p = 0.001$
for both tests). The diversity of the bacterial communities in the replicate samples of
the maritime chlorolichens *X. aureola* and *X. parietina* clustered according to sampling
location (Atlantic Ocean or Mediterranean Sea coast).

At the OTU level, the marine versus the maritime-inland terrestrial lichens were
associated with very different bacterial communities as illustrated in Fig. 4B, which shows
the 30 most abundant OTUs for either group. The closest relatives of the OTU representative
sequences assessed by *blastn* are presented in Table S4. Over 80% of the representative OTU
sequences showed >97% identity hits to previously deposited sequences. OTUs with a lower
% identity belonged mainly to the Bacteroidetes phylum. Interestingly, one Chloroflexi
OTU (OTU_12) that was recovered from *L. confinis* showed only a very low 91% identity
to its closest previously described relative (Table S4) and half of the closest relatives for the
marine lichen OTUs originated from marine intertidal outcrops (*Couradeau et al., 2017*).

A gammaproteobacterial OTU dominated in *L. pygmaea* and although this OTU was
also present in *L. confinis*, it was not the most abundant. *L. confinis* showed a more even
distribution of the OTUs compared to *L. pygmaea* as reflected by the diversity metrics.
Despite the relative spatial proximity of *X. aureola* and the *Lichina* lichens on the Atlantic
coast, there were very few shared OTUs among those genera. One OTU affiliated to the
candidate phylum FBP was present at low abundance in *L. confinis* and *Xanthoria* sp. but
not in *L. pygmaea*. The microbial communities recovered from the *L. pygmaea* samples

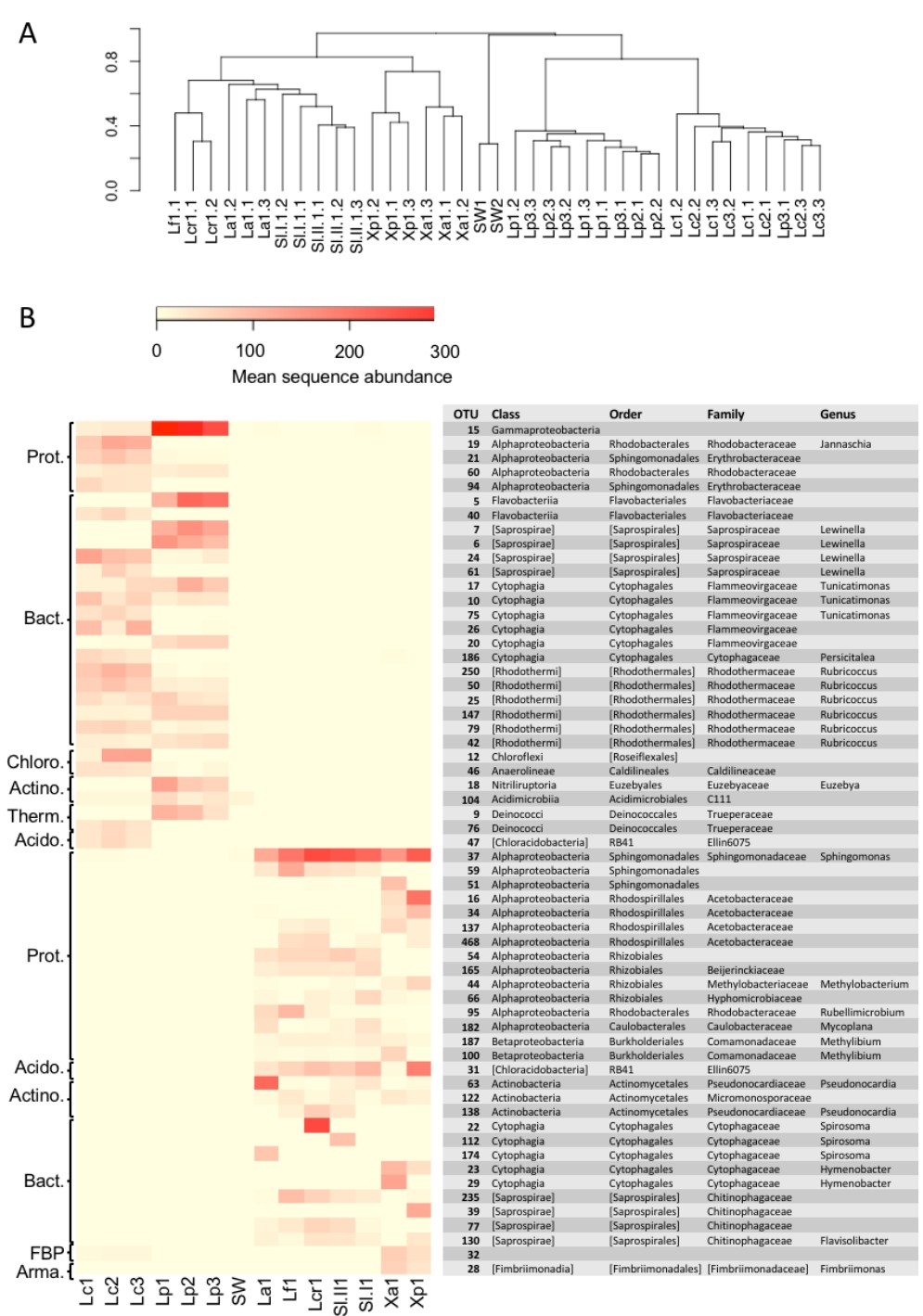

**Figure 4  Beta diversity of bacterial communities associated with the lichens or seawater assessed by Bray Curtis dissimilarity and average linkage clustering.** (A) Beta diversity of bacterial communities associated with the lichens or seawater assessed by Bray Curtis dissimilarity and average linkage clustering (B) Heatmap showing the relative abundance of the 30 most abundant OTUs among either the marine or the maritime-inland terrestrial lichens. The mean sequence abundance was calculated for each lichen sample cluster or the seawater samples. Lf, *L. fuscovirens*; Lcr, *L. cristatum*; La, *L. auriforme*; Sl, *S. lichenoides*; Xp, *X. parietina*; Xa, *X. aureola*; SW, seawater; Lc, *L. confinis*; Lp, *L. pygmaea*.

**Table 2** Levels of phylogenetic clustering of the lichen associated bacterial communities as measured by the Nearest Taxon Index (NTI) and the Net Relatedness Index (NRI).

| Lichen species | N ° taxa | NRI | runs > null | NTI | runs > null |
|---|---|---|---|---|---|
| *Coastal Brittany lichens* | | | | | |
| L. confinis | 271 | 1.42 | 905 | 2.30** | 991 |
| L. pygmaea | 281 | 3.49** | 998 | 1.68* | 951 |
| X. aureola | 224 | 3.13** | 997 | 2.15* | 978 |
| *Inland Austrian lichens* | | | | | |
| S. Lichenoides | 294 | 2.15* | 983 | 0.12 | 552 |
| L. auriforme | 260 | 2.53** | 992 | 1.73 | 944 |
| L. cristatum | 226 | 1.32 | 902 | 0.92 | 825 |
| L. fuscovirens | 205 | 1.05 | 859 | 1.61 | 938 |

**Notes.**

The values of each metric for the coastal or inland lichens are presented together with the number of runs where the observed NRI/NTI values were greater than those of the null model.

*$p < 0.05$.
**$p < 0.01$.

also showed very few OTUs in common with the seawater that was sampled in proximity to this lichen. This suggests that the *L. pygmaea* communities were specifically associated to the lichen and very different from the free-living seawater communities.

Interestingly the *Lichina* lichens were associated with several OTUs belonging to the Rhodothermaceae family (Bacteroidetes) and the Deinococcus-Thermus and Chloroflexi phyla of which some members are meso- or thermophilic, highly resistant to UV radiation (Deinococcus) and phototrophic (Chloroflexi). Although the *Lichina* communities showed a similar genus-level composition with closely related OTUs (eg *Lewinella*, *Tunicatimonas*, *Rubricoccus*) present in both lichens, these OTUs often showed an inverse pattern of abundance between the two species, where OTUs in genera present in both species tended to have higher counts in only one of the species (Fig. 4).

Phylogenetic analysis has already been used to show that environmental filtering rather than competitive effects structure marine bacterial communities (*Pontarp et al., 2012*). Similarly, to examine the potential ecological processes that could have influenced the community structure of the non-cyanobacteria bacteria in association with the lichen species, the level of phylogenetic clustering (indicative of environmental filtering) or over-dispersion (competitive effects) was estimated from a calculation of the Net Relatedness Index (NRI) and the Nearest Taxon Index (NTI; (*Webb, Ackerly & Kembel, 2008*). With the exception of the NRI for communities associated with *L. confinis*, there was evidence of phylogenetic clustering of the bacterial OTUs for the marine and maritime lichens as observed by a positive NRI and NTI (Table 2).

## Shared and specific bacterial OTUs in the marine and maritime lichens

To explore the degree of overlap between the marine and maritime lichen bacterial communities in more detail, the OTUs consistently shared between the species, and potentially indicative of a core microbiome were identified (subsequently referred to as

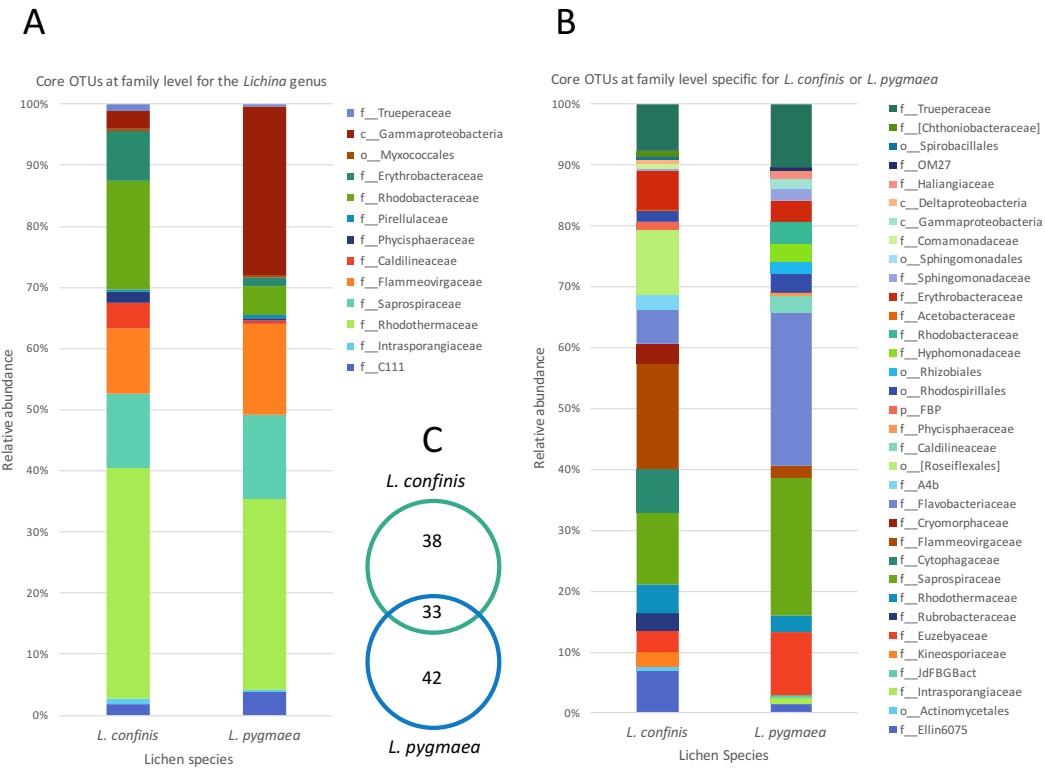

**Figure 5  Comparison of relative abundance of core or specific OTUs present in *L. confinis* and *L. pygmaea*.** Comparison of relative abundance of core OTUs present in (A) all replicates of both *L. confinis* and *L. pygmaea* or (B) the core OTUs specific for either species grouped at the family level. When family level assignment was not available, the next higher taxonomic level assigned was used. (C) Venn diagram showing the number of core OTUs shared or specific to *L. confinis* and *L. pygmaea*.

core OTUs). These OTUs were grouped at the family (or order when identification not possible) level and represented as relative abundance (Fig. 5). Setting a very conservative threshold of 100% presence in all replicates of the marine lichen species, only 33 OTUs were shared by both species (Fig. 5A). Of these, 15 were attributed to the Bacteroidetes families Saprospiraceae, Flammeovirgaceae and Rhodothermaceae (genera *Lewinella*, *Tunicatimonas* and *Rubricoccus* respectively). These OTUs accounted for around 60% of the sequences amongst the core OTUs and showed similar relative abundances between the two *Lichina* species (Fig. 5A). Other families included the Chloroflexi Caldilineaceae, and the Alphaproteobacteria families Rhodobacteraceae and Erythrobacteraceae that were more abundant in *L. confinis*, and a Gammaproteobacteria OTU in *L. pygmaea*.

When considering OTUs associated to each lichen species, 38 and 42 OTUs were specific (i.e., present at 100% of the replicates) for *L. confinis* and *L. pygmaea* respectively, and their relative abundances are shown in Figs. 5B and 5C, grouped at the family (or order when identification was not possible at the family level) level. The distribution and relative abundance of the family level were very different between the two species. For *L. confinis*, the most noticeable differences included a higher relative abundance of Acidobacteria

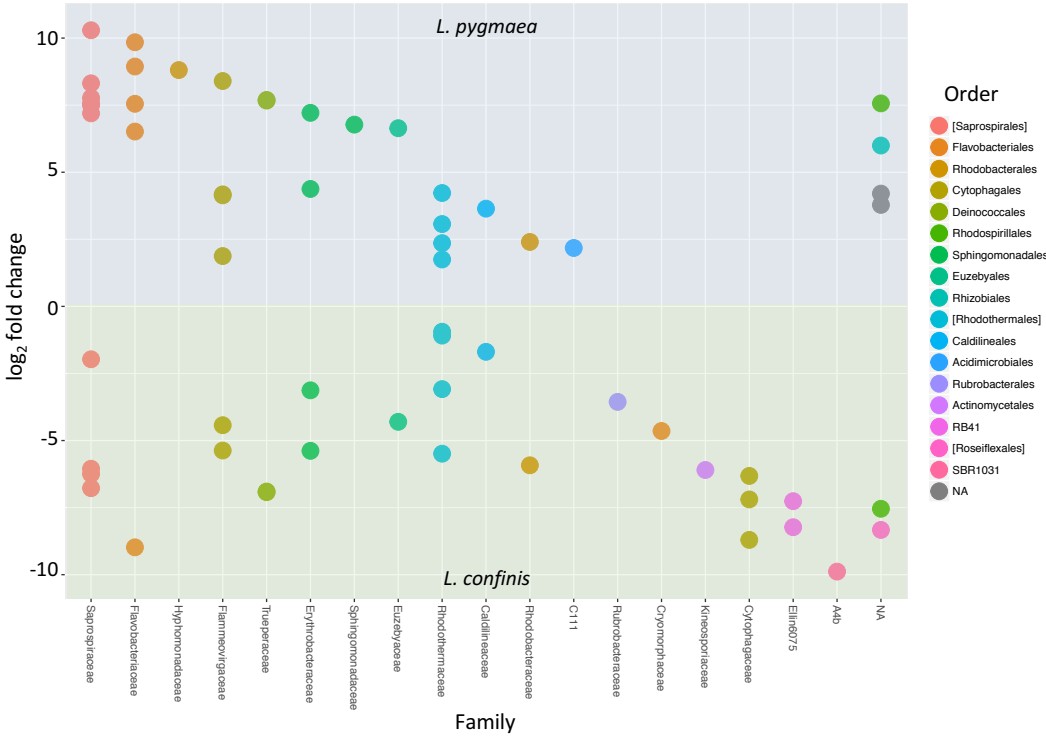

**Figure 6** **OTUs with significantly different relative abundances in *L.confinis* and *L. pygmaea*.** OTUs with significantly different relative abundances in *L.confinis* and *L. pygmaea* were assessed by the DESeq2 package using an adjusted *p* value < 0.001 and a base mean threshold > 20. NA : OTUs not assignable at the particular taxonomic level (order or family).

Ellin6075, Bacteroidetes Cytophagaceae and Flammeovirgaceae and the Chloroflexi order Rosiflexales. For *L. pygmaea*, the Actinobacteria family Euzebyaceae, and the Bacteroidetes families Saprospiraceae and Flavobacteriaceae dominated the specific core OTUs.

To determine the OTUs that showed statistically significant differential abundances between *L. confinis* and *L. pygmaea*, we used the DESeq2 negative binomial Wald test (*Love, Huber & Anders, 2014*). One third of the OTUs tested (120 out of 367 OTUs) showed a significantly different distribution between the two lichens (*P* < 0.001) and those OTUs with a mean abundance >20 are shown in Fig. 6. These OTUs were distributed in several shared families (left side of the plot) but we also observed differentially distributed OTUs that were distinct for each lichen. Interestingly, 29 out of the 33 core *Lichina* OTUs showed a statistically significant different distribution between the two species.

*X. aureola* collected above *L. confinis* on the littoral belt only shared one core OTU with the *Lichina* lichens that was assigned to the Gammaproteobacteria OTU, the most abundant OTU in *L. pygmaea*. *X. aureola* and *X. parietina* sampled from the Atlantic or Mediterranean French coasts shared 24 core OTUs (Fig. S3) that were dominated by the Proteobacteria (mostly Alpha- and Betaproteobacteria) accounting for over 50% of the sequences for each species. *X. parietina,* collected from the dryer Mediterranean coast,

showed higher relative abundances of Acidobacteria and the family Acetobacteraceae (Alphaproteobacteria) compared to *X. aureola*.

## Inland terrestrial cyanolichen-associated bacterial communities

Out of the four inland terrestrial lichen species, *L. auriforme* and *S. lichenoides* showed the highest diversity and species richness, followed by *L. cristatum* and *L. fuscovirens* (Fig. 2). As observed in previous surveys of terrestrial lichen bacterial diversity, these inland lichen species were also dominated by the phylum Proteobacteria, attributed mainly to the class Alphaproteobacteria (Fig. 3). The phylum distributions were generally similar amongst these species except the Bacteroidetes were almost twice as abundant in *L. cristatum* and *L. fuscovirens*, compared to the abundance in the other two species.

The Bray–Curtis dissimilarity dendrogram (Fig. 4A) shows that the bacterial communities associated with *L. auriforme* were closely related to those of *S. lichenoides* (both species were from the humid forest gorge) and this was confirmed with an ANOSIM test that showed no significant difference between the species groups ($R = 0.239$, $p = 0.064$). The other two species *L. cristatum* and *L. fuscovirens* from the dryer, rocky zone clustered apart and the separation of these two groups (humid forest versus dry/exposed) was also significant ($R = 0.582$, $p = 0.017$).

At the OTU level, the genus *Sphingomonas* (Alphaproteobacteria) was highly represented in the maritime and inland terrestrial lichens and remarkably absent in the marine lichens. Other OTUs present in all the inland terrestrial species included two OTUs assigned to the order Rhizobiales, a Betaproteobacteria OTU assigned as Methylibium, an Acidobacteria OTU and two Bacteroidetes OTUs belonging to the Chitinophagaceae. For the inland terrestrial lichens, significant NRI values were only observed for *S. lichenoides* and *L. auriforme* but none of these lichens showed significant phylogenetic clustering at the tree tips for the NTI (Table 2).

## Comparison of culturable species and total bacterial diversity from coastal lichens and *L. auriforme*

The 16S rRNA gene sequences of previously isolated strains originating from *L. confinis*, *L. pygmaea*, and *L. auriforme* were compared to the sequences in this study to investigate the representation of the culturable fraction relative to the bacterial diversity assessed by the cultivation-independent molecular method. Thirty-four strains out of the 254 previously isolated strains had 16S rRNA sequence identities >97% to the OTUs identified in this study (Table S5). The majority of the OTUs that had 16S rRNA sequences with high identity (>99%) to the strains were of low abundance (<0.5% of total sequences) in our study. One exception included the abundant OTU_37 (*Sphingomonas*; ~4–14% of total sequences) that shared 99% identity to a strain isolated from *L. auriforme*. One of the top 30 most abundant OTUs from the marine lichens (OTU_21, Erythrobacteraceae) shared 98% sequence identity with a strain isolated from *L. pygmaea*.

## Diversity of lichen photobionts and non-photobiont cyanobacteria

The major photobiont of each of the cyanolichens was identified as the cyanobacterial OTU with the highest relative abundance among all the cyanobacterial sequences, and

these identifications were confirmed with that reported in the literature (*Otálora et al., 2010*; *Ortiz-Álvarez et al., 2015*). The marine and inland terrestrial cyanolichens were associated with a different major photobiont. For *L. confinis* and *L. pygmaea*, the majority of the cyanobacterial reads clustered into a single *Rivularia* sp. OTU accounting for 51–82% of the reads for the *L. confinis* replicate samples and 25–79% for those of *L. pygmaea*. The remaining cyanobacterial sequences belonged principally to the order Chroococcales (including the family Xenococcaceae and the genus *Chroococcidiopsis*) and the family Pseudanabaenaceae (Fig. S4). For the inland terrestrial lichens, two abundant *Nostoc* OTUs were identified. The first OTU was attributed as the primary photobiont in *L. auriforme*, *L. cristatum* and *L. fuscovirens*, accounting for 92–98% of all cyanobacterial sequences and was also present (albeit at less than 5.5% of all cyanobacterial sequences) in *S. lichenoides*. The second *Nostoc* OTU was the major photobiont of *S. lichenoides*, accounting for 92–94% of total cyanobacterial sequences in four out of five replicates (67% in the fifth) but had a low abundance in the other terrestrial lichens (at less than 0.5%). Therefore, in contrast to the marine lichens, the inland terrestrial lichen photobionts dominated the cyanobacterial sequences. The non-photobiont OTUs of these lichens included, amongst others, the genus *Calothrix* and members of the Chroococcales and together they did not exceed 10% of the total cyanobacterial sequences (Fig. S4). As to be expected for chlorolichens, few cyanobacterial sequences (<5% of total reads) were obtained for the *Xanthoria* lichens but one chloroplast OTU was abundant in both species. Although only assigned as far as the order Chlorophyta, this OTU most likely represents the major photobiont of *Xanthoria* which is known to be the green alga *Trebouxia*.

## DISCUSSION

To date, the largest body of lichen research has been undeniably devoted to terrestrial species. We address this knowledge gap in applying, for the first time, next generation Illumina sequencing to characterise the bacterial communities associated to strict marine cyanolichens and a maritime chlorolichen, occupying different areas of the littoral zone. Whereas the two marine *Lichina* species, occurring in two neighbouring zones, share some common OTUs, many are specific to each species and there was very little overlap between the marine and maritime lichen bacterial communities.

### *Lichina* bacterial communities

A comparison of the *Lichina* bacterial community diversity allowed us to identify potential core OTUs and specific OTUs for each species, reflecting the potential adaptation of *L. confinis* or *L. pygmaea* to their different littoral zones. The *Lichina* bacterial communities shared 15 of 33 core OTUs that were attributed to only three Bacteroidetes genera, *Rubricoccus*, *Lewinella*, and *Tunicatimonas*, as well as several Alphaproteobacteria OTUs from the Erythrobacteraceae and Rhodobacteraceae families, and a Chloroflexi OTU. *Tunicatimonas pelagia* which belongs to the Flammeovirgaceae family was initially isolated from a sea anemone (*Yoon et al., 2012*) and other members of this family are known to associate with corals (*Apprill, Weber & Santoro, 2016*). *Lewinella* belongs to the Saprospiraceae family that was found as an epiphyte of red algae (*Miranda et al., 2013*) and

on filamentous bacteria identified as Chloroflexi (*Xia et al., 2008*), and isolates from this genus were also cultured from marine lichens (*Sigurbjornsdottir et al., 2014*; *Parrot et al., 2015*). Notably, the bacterial groups associated with the *Lichina* lichens were also similar to those found in the surface layer of hypersaline microbial mats (*Schneider et al., 2013*) and in epiphytic communities on the red macroalga *Porphyra umbilicalis* (*Miranda et al., 2013*). The co-occurrence of these different bacteria in microbial mats and on various marine hosts suggests that they have adapted to similar environmental conditions. In functioning as a consortium, these communities could potentially increase their robustness and that of their hosts by expanding their range of metabolic capacities and resistance mechanisms to stress (*Hays et al., 2015*).

Interestingly there were multiple *Lewinella*, *Tunicatimonas* and *Rubricoccus* OTUs in each lichen species and OTUs that were affiliated to the same genus were differentially distributed between the two *Lichina* species (e.g., see the *Lewinella* OTUs 6, 7, 24 and 61 in Fig. 4). This could suggest that these bacterial genera belong to ecologically differentiated pools as was already shown for the for the *Rivularia* cyanobionts of the same lichen species (*Ortiz-Álvarez et al., 2015*). This mechanism of niche adaptation is thought to occur in extreme environments and could also apply to the bacteria associated to these marine lichens, which became adapted to the intertidal or supratidal zones. The presence of these closely related clusters was also supported by the positive and significant NTI values (*Webb et al., 2002*) indicative of environmental filtering effects on the *Lichina* bacterial communities.

Rocky tidal zones are thought to be one of the most stressful habitats on earth (*Miller, Harley & Denny, 2009*) and both marine lichen species bacterial communities were associated with several groups of bacteria that are known to be heat- or radiation-resistant. These include several OTUs identified as core OTUs that were affiliated to the genus *Rubricoccus* of the Rhodothermaceae (*Park et al., 2011*), to the phylum Chloroflexi (that contains moderately thermophilic/thermophilic groups), and to the family Truperaceae in which the isolated species are thermophilic and nearly as radiation resistant as *Deinococcus* (*Albuquerque et al., 2005*).

Despite sharing 33 core OTUs, the two marine *Lichina* species *L. confinis* and *L. pygmaea* were associated with significantly different bacterial communities at the OTU level and remarkably the Bray Curtis dissimilarity distance between these communities was greater than that of the distance between the communities associated with two different inland terrestrial cyanolichen genera, *Lathagrium auriforme* and *Scytinium lichenoides* (Fig. 4A). This difference was also reflected in the high number of OTUs (>120) that displayed a significantly different abundance between the two species (Fig. 6). These significant differences were despite the relatively short distance that separated the species on the vertical gradient (height on the littoral belt). Other studies show evidence that height on the littoral zone is the most important factor explaining the rocky coastline species variance with slope, exposure, substrate and orientation explaining less (*Chappuis et al., 2014*). *L. confinis* occurs higher up on the littoral belt, being mostly exposed to the air and would be subject to higher temperatures and UV radiation. This could explain why some thermophilic groups such as Chloroflexi and the Actinobacterial genus *Rubrobacter* that

include thermophilic species resistant to radiation (*Ferreira et al., 1999*) were four-fold and 13-fold respectively more abundant in *L. confinis* than *L. pygmaea*. The bacterial groups that were more abundant in *L. pygmaea* when compared to *L. confinis* included the Gammaproteobacteria OTU related to a seagrass epiphyte *Granulosicoccus* (*Kurilenko et al., 2010*) an OTU affiliated to the Actinobacterial genus *Euzebya*, recovered from a sea cucumber (*Kurahashi et al., 2010*) and several Flavobacteriaceae and Rhodobacteraceae OTUs. Marine epiphytic Actinobacteria, and *Phaeobacter* (Rhodobacteraceae), associated with a variety of marine plants and animals, are known to produce bioactive molecules (*Rao et al., 2007*; *Valliappan, Sun & Li, 2014*). Such molecules have been shown to inhibit settlement of algal spores, fungi and larvae and may contribute to the host's defence against biofouling (*Rao et al., 2007*). As *L. pygmaea* is immersed for half the day, it may be more subject to biofouling than *L. confinis* and hence the biofilm communities covering the *L. pygmaea* may play an important role in the inhibition of attachment of undesirable organisms.

Despite the significantly different bacterial communities observed in the two vertically segregated *Lichina* species in the littoral belt, we did not find clear evidence for significant differences between the samples of each species across horizontal distances (minimum 6 m and maximum 37 m between clusters). This was indicated by beta-diversity analyses both on the whole or filtered dataset which included the most abundant OTUs and which revealed a lack of clustering of the replicates according to site. A lack of geographical influence on the lichen associated bacterial communities was also observed in other studies where sampling was carried out on a small spatial scale (*Grube et al., 2009*; *Bates et al., 2011*). In a study where lichen sampling was carried out at a much greater spatial scale, across the North American continent, a significant effect of geographical location on bacterial communities associated with a specific species was observed (*Hodkinson et al., 2012*).

## Low culturability of marine lichen associated bacteria

Culturing of bacterial isolates from novel environments is essential for natural product discovery. The harsh environments, in which the *Lichina* lichen species are found, could provide a wealth of bioactive products, particularly from the potentially thermophilic/radiation resistant bacteria identified in our study. However, only 34 OTUs showed significant similarity to the 254 previously isolated cultured isolates (*Parrot et al., 2015*) even when lowering the similarity threshold to 97%. Moreover, three OTUs that were 100% identical to isolates were present at low abundances in the Illumina dataset. Whereas the Bacteroidetes phylum accounted for around 53% of the 16S rRNA sequences from the marine lichens, only four strains from this phylum were cultured. These results can be partly explained by the culturing strategy which was intentionally skewed towards the isolation of *Actinobacteria* by using Actinobacteria Isolation Agar (*Parrot et al., 2015*). As the bacterial isolates were obtained from lichens sampled 12–18 months prior to those that were analysed in this study (even though they were collected from the same site), we cannot rule out the influence of different sampling times on the bacterial composition. Nevertheless, the low culturability of bacteria from certain environments is well known

(*Amann, Ludwig & Schleifer, 1995*) and our results further highlight the importance of parallel culture independent molecular methods to gain a more complete picture of the microbial diversity. Such results, combined with metagenomic approaches to target gene clusters of interest (*Vester, Glaring & Stougaard, 2015*) and novel isolation strategies incorporating lichen extracts into culture media (*Biosca et al., 2016*) should improve the bioprospecting of marine lichens. Nevertheless, the fact that some of the previously cultured strains were relatively close to the community described by our study, suggests that some of the bacteria associated with lichens might not be as recalcitrant to culture when compared to bacteria from other environments, highlighting the interest of these associations as sources of cultured microbial diversity.

## Marine versus terrestrial lichens: challenging the paradigm of Alphaproteobacterial dominance

Here we show that bacterial communities in two strictly marine lichen species were completely different from those of two maritime and four inland terrestrial lichen species and that this difference is immediately visible at the phylum level with 50% of the sequences assigned to Bacteroidetes in *Lichina*, with a lower proportion of Proteobacteria (20–30%), the converse of the inland lichens from the humid wooded zone (15–19% Bacteroidetes and 50% Proteobacteria). At the OTU level, the differences were even more marked, with no overlap between the marine and inland terrestrial cyanolichens studied. The *X. aureola* communities (sampled above *L. confinis* on the rocky shore) only shared a single OTU with the *Lichina* lichens (Gammaproteobacteria OTU_15) which was only present at low abundance (0.05%). High abundances of Bacteroidetes were also observed in a different marine lichen *Hydropunctaria maura* (*Bjelland et al., 2011*) and supports the idea that this group of bacteria are major components of the bacterial communities associated with marine lichens, whereas Alphaproteobacteria are well confirmed by several studies as the dominant group associated with terrestrial lichens (*Cardinale et al., 2008*; *Grube et al., 2009*; *Bates et al., 2011*; *Mushegian et al., 2011*; *Hodkinson et al., 2012*). The marine and the maritime/inland terrestrial lichen mycobionts did belong to two different classes (Lichinomycetes and Lecanoromycetes respectively; see Table 1) so we cannot rule out that this significant mycobiont taxonomic separation could be partly responsible for the difference in bacterial community composition observed. Nevertheless, a survey of bacterial communities associated with different cyanolichens and chlorolichens showed that taxonomic dissimilarity of the mycobiont did not necessarily correlate with bacterial community dissimilarity (*Hodkinson et al., 2012*). In that study, although *Dictyonema* belongs to the division Basidiomycota while the other lichens belonged to the Ascomycota, the bacterial communities of *Dictyonema* were still composed of >50% Alphaproteobacteria distributed into the same four orders as for the majority of the other lichens. To be able to separate more clearly the influences of lichen taxonomy and habitat on the diversity of the associated bacterial symbionts, ideally, a terrestrial *Lichina* lichen species or another lichen belonging to the Lichinaceae family should be characterised in the future.

In marine environments Bacteroidetes are specialised in degrading polymeric organic matter and are adapted for an attached lifestyle by the production of adhesion proteins

(*Fernández-Gómez et al., 2013*). Although Bacteroidetes are often reported attached to particles they are also known to be epiphytic on a variety of marine algae from temperate regions and often co-occur with Proteobacteria (*Lachnit et al., 2011*; *Burke et al., 2011*; *Wahl et al., 2012*; *Miranda et al., 2013*). Although high abundances of Alphaproteobacteria sequences were detected on the green alga *Ulva australis* from Botany Bay NSW Australia (*Burke et al., 2011*), Bacteroidetes was the most abundant group associated with the red alga *Porphyra umbilicalis* (*Miranda et al., 2013*) in Maine, US and with red and brown algae sampled during summer months in Kiel Bight, Germany (*Lachnit et al., 2011*). This suggests that Bacteroidetes may be important members of marine epiphytic and perhaps epilichenic communities, although this would need to be confirmed by microscopy-based (FISH) approaches.

Other differences between the marine and inland terrestrial lichens included the relative abundance of the bacterial (non-cyanobacterial) sequences compared to the cyanobacterial sequences that were as low as 20% in *L. auriforme* but accounted for 50–60% in the marine lichens (Fig. S1). The marine lichen *H. maura* also had a higher estimated concentration of bacterial cells in the lichen thallus compared to inland lichens (*Bjelland et al., 2011*). These observations suggest that bacteria associated with marine lichens might play a greater role in the functioning of the lichen symbiosis but this would need to be confirmed by microscopy analyses and for example, metabolite profiling.

Several roles of lichen-associated bacteria have been suggested and include nutrient supply (e.g., by nitrogen fixation and phosphate scavenging), nutrient recycling, protection against UV radiation, resistance to abiotic and biotic environmental stressors, and thallus degradation (*Eymann et al., 2017*). Unlike terrestrial lichens, marine lichens would need to be adapted to much more variable gradients of UV light, salinity (osmotic) stress, heat, desiccation and mechanical stresses from wave action and therefore we would expect the associated bacteria to play a role in resisting these particularly variable conditions. Although marine lichens can provide their own UV protection by the production of a parietin derivative in the case of *Xanthoria* and mycosporine-like amino acids in the case of the *Lichina* sp. (Roullier et al., 2011), it is possible that the bacteria that colonise the surface of the lichen like a biofilm (*Cardinale et al., 2012b*) could provide a protective layer conferred by other photoprotective pigments. The majority of the identified OTUs were affiliated to pigmented bacterial families, and several potentially phototrophic bacterial groups such as the aerobic anoxygenic phototropic (AAP; e.g., Chloroflexi, Erythrobacteraceae, Rhodobacteraceae) or rhodopsin-containing bacteria (e.g., Flavobacteriaceae, (*Yoshizawa et al., 2014*), Rhodothermaceae (*Vavourakis et al., 2016*), Deinococcus-Thermus) that contain carotenoids that may have a photoprotective role.

To alleviate salt stress and resist desiccation, marine lichens and their photobionts accumulate compatible solutes that are mainly sugars, sugar alcohols and complex sugars, lowering the water potential and hence reducing water movement out of the cells (*Delmail et al., 2013*). Indeed *L. pygmaea* which is immersed in seawater for half the day was shown to resist salt stress better than *Xanthoria aureola*, which is located higher up on the littoral belt and only receives sea spray (*Delmail et al., 2013*). Marine biofilms are considered to be analogous to a multicellular organism in that it functions like a "second skin", replacing

the hosts top layers as the new interface between the host and the environment (*Wahl et al., 2012*). Such a function could be imagined for the bacteria associated with *L. pygmaea* that could contribute to the osmotic adaptation of the lichen by maintaining a layer of cells in osmotic balance with the surrounding water. This could be achieved by different mechanisms including the synthesis and uptake of organic osmolytes and by using a sodium pumping rhodopsin as inferred from Bacteroidetes metagenomes recovered from hypersaline lakes (*Vavourakis et al., 2016*). While this is speculative, bacterial communities inhabiting the rhizosphere of coastal plants were shown to confer salt stress acclimatisation to the plants, potentially though the use of ion and compatible solute transporters (*Yuan et al., 2016*).

In cyanolichens or tripartite lichens, fixed carbon and nitrogen are supplied to the mycobiont by the photobiont whereas phosphate was shown to be a limiting macronutrient since cyanolichen growth was stimulated after immersing the thalli in a phosphate solution (*McCune & Caldwell, 2009*). This was also supported by metagenomic and proteomic analysis of two terrestrial lichens that attributed an important role of phosphate solubiliation genes to the associated microbial community, more specifically to the Alphaproteobacteria (*Grube et al., 2015*; *Sigurbjornsdottir, Andresson & Vilhelmsson, 2015*). The *Lichina* lichens may be less subject to phosphate limitation because *L. pygmaea* is immersed in seawater for half a day and *L. confinis* receives sea spray in a coastal area that is not particularly phosphorus limited. Nevertheless, some of the associated bacteria may play a role in phosphate uptake from the seawater or in the recycling and transfer of phosphate from senescing parts of the thallus to the growing apices as was shown for *Cladonia* (*Hyvärinen & Crittenden, 2000*). Such nutrient recycling may be carried out in part by the Bacteroidetes that are known to produce a wide range of enzymes capable of degrading polymeric organic matter including glycoside hydrolases and peptidases (*Fernández-Gómez et al., 2013*).

## CONCLUSIONS

Here we show the striking differences between bacterial communities associated with marine and terrestrial lichens and even between marine lichens inhabiting different littoral zones. This represents a first piece in the puzzle in understanding the role of the associated bacteria in the marine lichen symbiosis, which is lagging much behind that of the terrestrial lichens. To begin to elucidate the functions of these communities, future work should include a multidisciplinary approach combining microscopy techniques to determine the spatial organisation of the core microbiome members and "omic" approaches to understand their functional role. An integrative study using metagenomics, metatranscriptomics and metabolomics could be performed on *L. pygmaea* during high and low tide to reveal a wealth of information on how the symbiotic partners interact and adapt to these extreme conditions. Moreover, such information would not only further our understanding of the marine lichen symbiosis but could also guide future bioprospecting efforts.

## ACKNOWLEDGEMENTS

We thank the BIO2MAR platform (http://bio2mar.obs-banyuls.fr) for providing technical support and access to instrumentation. The following reagent was obtained through BEI Resources, NIAID, NIH as part of the Human Microbiome Project: Genomic DNA from Microbial Mock Community B (Even, Low Concentration), v5.1L, for 16S rRNA Gene Sequencing, HM-782D. We would also like to thank Dr. Nicolas Magain for his thorough and thoughtful review, particularly on the lichen taxonomy, which greatly helped to improve this paper.

### Funding

This study was funded by by the project MALICA (10-INBS-02-01) of the Agence Nationale de la Recherche, France and the INSA Rennes for the ministerial grant for the PhD of Delphine Parrot. The funders had no role in study design, data collection and analysis, decision to publish, or preparation of the manuscript.

### Grant Disclosures

The following grant information was disclosed by the authors:
MALICA: 10-INBS-02-01.
INSA Rennes.

### Competing Interests

Marcelino T. Suzuki is an Academic Editor for PeerJ.

### Author Contributions

- Nyree J. West conceived and designed the experiments, performed the experiments, analyzed the data, contributed reagents/materials/analysis tools, prepared figures and/or tables, authored or reviewed drafts of the paper, approved the final draft.
- Delphine Parrot conceived and designed the experiments, performed the experiments, contributed reagents/materials/analysis tools, prepared figures and/or tables, authored or reviewed drafts of the paper, approved the final draft.
- Claire Fayet performed the experiments, authored or reviewed drafts of the paper, approved the final draft.
- Martin Grube and Sophie Tomasi conceived and designed the experiments, performed the experiments, contributed reagents/materials/analysis tools, authored or reviewed drafts of the paper, approved the final draft.
- Marcelino T. Suzuki conceived and designed the experiments, contributed reagents/materials/analysis tools, authored or reviewed drafts of the paper, approved the final draft.

## DNA Deposition

The following information was supplied regarding the deposition of DNA sequences:

Sequence data is available at the European Nucleotide Archive under the study number: PRJEB23513.

## Data Availability

West, Nyree; Parrot, Delphine; Fayet, Claire; Grube, Martin; Tomasi, Sophie; Suzuki, Marcelino T. (2018): Lichen associated bacteria 16S rRNA Illumina sequences. figshare. Fileset. https://doi.org/10.6084/m9.figshare.5891509.v1.

## Supplemental Information

Supplemental information for this article can be found online at http://dx.doi.org/10.7717/peerj.5208#supplemental-information.

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
