# Peer review of "Marine cyanolichens from different littoral zones are associated with distinct bacterial communities"

_PeerJ, doi:10.7717/peerj.5208_

## Round 0.1 · original submission · Minor Revisions

I agree with reviewers that this is a well written study. I appreciate the novelty of the topic and the results provide outstanding insight on lichen biology.

As reported by reviewers, some more detail could be added to further emphasize the phylogenetic difference between the two Lichina species and all the other lichens studied. I understand that there are several pragmatic reasons for species selection, but this bias should be clearly explained in the discussion, in order to make pros and cons of the research more clear for the audience.

·

Basic reporting

No comment

Experimental design

No comment

Validity of the findings

This is an overall interesting study. The paper is well written and the sections are clearly organized and presented. The amount of data and results is good.

The result that “real marine” lichens (here two species of Lichina) have a microbiome that is more distinct from the “semi marine” Xanthoria, than "semi marine" Xanthoria to terrestrial Collemataceae is interesting. The comparison of the bacterial communities between the different species is interesting, and the high abundance of thermophilic/radiation-resistant species in the marine lichens is interesting and could help explain their resistance to quite extreme and unusual conditions. This result might be the highlight of the paper. It is also interesting that the microbiome of marine lichens may show more similarity to the microbiomes of marine algae than terrestrial lichens.

However, the aim of the article is to compare marine lichens' microbiomes to semi- and non-marine lichens', which brings to what is, to me, the main weakness of the study: all marine lichens tested in this study are members of the class Lichinomycetes, whereas all other tested lichens belong to the Lecanoromycetes (they are in fact very distantly related), and moreover all terrestrial lichens belong to the same family (Collemataceae). It is therefore difficult to tease apart the influence of the ecology versus the influence of the taxonomy.

The sampling design doesn’t help to tease taxonomy and ecology apart, because all the Lichina were collected in a single region (whereas Xanthoria was sampled in two different areas; it is a little strange to sample the species that are the main focus of the paper less than others).

This doesn’t mean that the study is not worthy of publication, especially because fixing this problem would require to start from scratch, but this bias should be clearly explained in the discussion. It is unfortunate that taxonomic relationships in the paper are either ignored (the Lichinomycetes vs, Lecanoromycetes situation is never discussed) or wrongly inferred (as for the relationships inside Collemataceae).
* * *
Indeed, the authors don’t take into consideration the revision and actual classification of Collemataceae (three of the four species sampled have been transferred to new genera), which led them to wrong assumptions on the relationships between the four Collemataceae sampled (the three “Collema” are not more closely related to each other than to “Leptogium” lichenoides)

The authors need to read the paper by Otalora et al. 2013 on the phylogeny of the Collemataceae “A revised generic classification of the jelly lichens, Collemataceae” published in Fungal Diversity.

In fact, Collema auriforme is now Lathagrium auriforme. Collema crispum is now Blennothallia crispa, Leptogium lichenoides is now Scytinium lichenoides.

As a consequence, the sentence “The bacterial communities associated with C. auriforme were actually more closely related to those of L. lichenoides than the other two species of the same genus, C. crispum and C. flaccidum that formed a separate cluster.” doesn’t actually make sense because Collema and Leptogium in the sense they are used in this current form were two polyphyletic genera.

Therefore, C. crispum (now Blennothallia crispa), C. flacidum (still a Collema s. str.) and C. auriforme (now Lathagrium auriforme) are not more closely related to each other than to L. lichenoides. I suggest removing the part comparing the three Collema vs. Leptogium, or changing it to reflect the actual phylogenetic relationships. The two most closely related species are actually Scytinium lichenoides and Lathagrium auriforme, which is in agreement with your results. They then share a common ancestor with Blenothallia crispa, whereas Collema flaccidum is the least closely related taxon.

By the way, it would be good to know how those lichens were identified. Was their identity checked with ITS or were they identified solely based on morphology?

It would also be good to know to which species of Xanthoria the collected thalli belong.
* * *
Methods and analyses are adequate for the hypotheses tested. The amount of contamination in the Miseq run (0.5% of contamination) is concerning but the authors seem to have taken much care in processing the data. The number of species tested is low, but the results are interesting nevertheless.

When the authors discuss the relative abundance of the OTUs, it should be emphasized that the sensitivity and specificity to primers, the quality of the DNA and other factors can favor the amplification of some groups over others. For example, the discussion in lines 572-579 doesn’t take into consideration the fact that this could be a product of the sequencing itself, rather than an actually difference in the thallus, as those lichens have different photobionts and different bacterial communities that may be differentially sequenced by the method.

The authors go very fast on the cyanobacterial result. It is not clear if it is for the marine or the terrestrial lichens that up to 10% of cyanobacterial sequences were not from the photobiont. The number for the other category should be indicated as well. Authors should also say something about the sequence diversity for the photobiont themselves (only one genotype? Several?) and discuss the genera that was found in the non-photobiont sequences.

In the section “Comparison to previously cultured strains from coastal lichens and C. auriforme”, authors should give more information about the specimens from which previous culture-based methods were applied. For example, do they come from the same locality? We should know relevant information about those other specimens without the need of reading other papers. Also, why was Roccella included in the comparison if it was not sequenced in this study?

Additional comments

Abstract: Illumina should be capitalized

References are not formatted the same way: for examples references in lines 49, 55 and 56 have three different formats

Line 260: the authors go from a description of the results on the mock sample (up to line 259) to the description of other samples in line 260, but don’t say clearly what they are describing. Is it a total number of OTUs for everything? Does it include the mock sample? Is it only for the Lichina? The part between lines 260-270 needs to explain more carefully to which sequencing those results correspond.

Line 271-274. The authors define many terms that are confusing if the reader doesn’t go back to this paragraph every time. For example it is not intuitive that inland lichens and terrestrial lichens are different concepts. Is it really needed to use those terms? I feel these categories are not needed (they are not often used in the rest of the text).

Line 42: "to" is repeated twice, it should be “to which they are subjected daily”
Line 45: the word already is very oddly placed here
Line 49 needs a comma after the reference
Line 241: “terrestrial lichens were subject of blastn queries” I think should be “were subjected to blastn queries”
Line 326 show should be shows
Line 330: at least one comma is needed after sequences, but I would suggest splitting these long sentences in a few distinct ones.
Line 370 when the authors say “the OTUs consistently shared within the genus” but are only comparing two species in the genus, they can say “consistently shared between the two species” but cannot extrapolate to the entire genus.

Line 399 and around: What is called a core OTU should be clearly defined. What a dominant OTU is should also be defined.

Line 457: this communities should be these communities

Sometimes there is no space between the genus and species e.g., L.confinis in the caption of figure 6, sometimes Lichina is not italicized, please check carefully through the manuscript.

Reviewer 2 ·

Basic reporting

No comment

Experimental design

No comment

Validity of the findings

No comment.

Additional comments

Interesting and important study!

As there are so many aspects to cover it is difficult to refine these types of manuscripts into an informative and easily understandable form - I think the authors have done a fine job!

One could always argue for adding info on this and that issue but that would necessarily increase overall length.

However, maybe some more detail could be added to further emphasize the phylogenetic difference between the two Lichina species and all the other lichens studied (lines 89-95). One can argue that to more specifically elucidate the relative importance of habitat factors versus possible phylogenetic influence the two marine lichens should preferably have been compared with some inland species of Lichinaceae (Lichinales) and not with species representing other orders of Ascomycota?

Also the possible influence of thallus morphology (surface/volume ratios, gelatinous thalli versus air-filled medulla, etc.) could be touched upon - but as said, one cannot discuss everything in one paper.

Line 274 Replace period before ”inland lichens” with comma.
Line 316 This is in contrast to the…
Lines 369-372 Consider rephrasing this text (maybe by splitting into several sentences).

Reviewer 3 ·

Basic reporting

This article is very well written. The authors did a great job in clearly articulating the goals of the study. They also were thoughtful when referencing the literature.

Regarding the organization of the text, it would be helpful if the authors listed their questions on page 5 lines 123-128 in the same order that the questions were answered.

Regarding figure 2, the color choice makes it difficult to distinguish between the samples. For example, both on the screen and on paper, it was difficult to separate L confinis from L. Lichenoids. Likewise L. pygmaea and seawater were hard to separate by color.

Page 15, First paragraph. Please include information about the symbiont of Xanthoria.

Experimental design

The question of marine lichens microbiome is well defined and very interesting. It has not been addressed before this study and is relevant both from an ecological point of view as well as the potential for bioprospecting. The methods were well described.

The authors are very careful and have gone so far as to include a mock community in the sequence analysis. They candidly reported that the mock community showed some contamination (~0.5%). To account for contamination, the authors removed sequences with low variance across their samples. While ideally they would resequence their samples, this treatment is acceptable. However, the authors should mention somewhere in the text that it could introduce a bias, particularly with low abundance sequences since not all sequences are amplified equally during the sequencing reactions.

Minor comments:
1) Page 6 line 135. It would be helpful to cite in the text the how far apart the clusters are from each other (it is currently in the supplementary material) but from reading the text it is hard to know if they were miles or meters apart. It would also be helpful to clarify the distances in page 18 line 527 when the authors say there was no effect across horizontal distances.
2) Page 6 Line 143. Please indicate what part of the lichen was sampled

Validity of the findings

Overall, the conclusions are supported by the data.

My only concern is the comparison of the cultured samples to the culture independent results. Ideally the cultured samples would have been freshly cultured from the same lichen isolates used for the culture-independent study but I realize that is not possible at this point. Were the cultured samples taken from the same site? The data in the paper shows, at least with Xanthoria, the location affects the composition. The description of these results should be expanded. Did the host species correlate in the samples?

Overall the discussion was excellent. The most interesting point was the comparison of terrestrial versus marine lichens. This could have been put earlier to emphasize it more. (it was somewhat buried in the discussion).

Minor comments:
1. Page 14 Line 418-420, was the difference between the different Xanthoria samples calculated by DESeq2 or by the OTUS specific for each location?

2. Page 21 lines 635 to 651. The last paragraph of the discussion is very speculative. While it is within the guidelines of the journal to include some speculation in the discussion, it is a bit distracting and could be shortened.

---

## Round 0.2 · Minor Revisions

The revised version of the paper is considerably improved. The reviewer and I suggest just a few very minor corrections...

·

Basic reporting

This is a much better version of the manuscript. As all my comments have been taken into consideration, I think the paper is now suitable for publication. I would however recommend one more careful read to improve the writing of the text, especially in the newly-written parts. This would increase the quality of the paper and make it a nicer read.

A few suggestions:
In the abstract, microbiome has been replaced by microbiont in the first sentence. I’m not familiar with this term and it adds confusion. Microbial or microbiome would better fit.
Line 33: I would say “strictly marine”
Line 58: I think a word is missing, maybe “and”, before allows
Line 61: I don’t think you should say “also” as you haven’t given examples yet.
Line 64: I’d rather say “fit with” than “fit to”
Line 68: the reference shouldn’t be inside parentheses. Same for line 140.
Line 143 lead should be led
Line 148-150 you wrote characterised then characterized
Line 151: associated to should be associated with
Line 171 were should be was
Line 544 none this should be none of this

Experimental design

The experimental design is sound and my comments have been taken into consideration.

Validity of the findings

Findings are valid.

---

## Round 0.3 · accepted · Accept

Very nice paper! Thank you for submitting to PeerJ.

#